# Critical Factors for In Vivo Measurements of Human Skin by Terahertz Attenuated Total Reflection Spectroscopy

**DOI:** 10.3390/s20154256

**Published:** 2020-07-30

**Authors:** Lixia Wang, Sayon Guilavogui, Henghui Yin, Yiping Wu, Xiaofei Zang, Jingya Xie, Li Ding, Lin Chen

**Affiliations:** 1Shanghai Key Lab of Modern Optical System, Terahertz Spectrum and Imaging Technology Cooperative Innovation Center, Terahertz Technology Innovation Research Institute, University of Shanghai for Science and Technology, Shanghai 200093, China; 182550402@st.usst.edu.cn (L.W.); sayonguilavogui518@yahoo.com (S.G.); 18721299257@139.com (H.Y.); 1812440306@st.usst.edu.cn (Y.W.); xfzang@usst.edu.cn (X.Z.); xiejy@usst.edu.cn (J.X.); sunnylding@163.com (L.D.); 2Shanghai Institute of Intelligent Science and Technology, Tongji University, Shanghai 200092, China

**Keywords:** terahertz time-domain spectroscopy, attenuated total reflection, human skin, in vivo

## Abstract

Attenuated total reflection (ATR) geometry is a suitable choice for in vivo measurements of human skin due to the deep penetration of the field into the sample and since it makes it easy to measure the reference spectrum. On the other hand, there are several critical factors that may affect the terahertz (THz) response in these kinds of experiments. Here, we analyse in detail the influence of the following factors: the contact positions between the thumb and the prism, the contact pressure, the contact duration, and the materials of the prism. Furthermore, we use the THz-ATR technology to evaluate different types of handcream and also establish the theoretical model to investigate the reflectivity after interacting with the skin. The results agree well with experimental ones. Our analysis makes it clear the importance of controlling the above factors during measurements to enable reliable THz response and results which, in turn, may be used to monitor water motion in human skin and to predict possible diseases.

## 1. Introduction

Terahertz (THz) radiation (0.1 to 10 THz) has attracted a lot of attention in diverse fields [1,2,3,4,5,6,7,8,9,10,11,12,13,14,15,16,17,18,19]. Following the increase in the number of diabetic patients, attention has been devoted to the prevention and treatment of diabetes. In this framework, terahertz (THz) technology has been exploited for monitoring the human skin to assess blood glucose levels. Although human skin detection has been conducted in the visible and near infrared, THz detection is considered as a promising detection technique developed, which has various advantages, such as the high sensitivity to water, nonionising features and low photon energy [20]. Pickwell et al. investigated the interaction of pulsed THz radiation with normal human skin on the forearm and palm of the hand in vivo [21], whereas Bryan et al. demonstrated the application of terahertz pulse imaging for the in-vivo study of human tissue [22]. Gusev et al. studied optical properties of in vivo human skin using THz time domain spectroscopy [23]. These works provide a basis for the development of non-invasive reflective spectroscopic techniques for glucose monitoring and imaging. Sun et al. proved that using a reflection geometry for in vivo THz imaging of humans [24,25], an increase of the contact duration and pressure correspond to a reduction of the amplitude of measured THz signal, and to a small increase of the phase. Truong et al. aimed to analyse phantoms mimicking breast tissue using THz imaging in reflection [26]. Fan et al. demonstrated the capability of terahertz imaging to quantitatively measure subtle changes in skin properties in reflection mode [27]. In turn, compared to transmission and reflection geometry, experiments using attenuation total reflection (ATR) configuration offer several advantages [19,28,29,30,31,32,33,34], including (i) a larger amplitude of the reflected signal, which is less affected by strong absorption by water; (ii) a larger penetration depth of the field in the sample (the penetration depth is linear to the terahertz wavelength. The penetration depth of 0.1 THz is up to 192.1 μm) [35]; (iii) an easier measurement of the reference spectrum. Ogawa Yuichi et al. proposed the measurement of water content in human skin by using the THz ATR method [29]. Cherkasova et al. measured human skin spectra in vivo using THz time-domain spectroscopy (TDS) and an ATR optical scheme [31]. Later, they observed changes in the optical properties of human skin, correlated with changes in the blood glucose level [32,36,37,38]. Those analyses, however, do not take into account the other variables involved in the skin-prism contact process, and do not provide elements to assess their effects. In order to pave the way to potential applications of in vivo THz measurements of human skin in treatments, we need to assess how and to which extent the skin-prism contact process influences the THz properties. In this paper, we are going to analyse the following relevant factors: the contact positions between the thumb and prism, the contact pressure, the contact duration, and the material of the prism in the ATR system. Our experimental results demonstrate the importance of controlling variables during measurements. Furthermore, variations in skin optical properties have proved to correlate with changes in the blood glucose level [31,39]. We expect that our results may be used to monitor water motion in human skin and to predict possible diseases.

## 2. Experimental Methodology, Results and Discussions

### 2.1. Experimental Setup

The spectrometer employed for the in vivo investigation of samples is configured to perform reflection measurements. Experiments are performed using a combination of terahertz time domain spectroscopy (THz-TDS), described in detail in [40,41], and ATR [42,43]. The experimental setup has a frequency resolution of 9 GHz, which can fully describe a detailed spectrum of skin. According to the ATR principle, a triangular prism was designed, as shown in Figure 1.

In order to control the pressure of the human in contact with the prism surface, a digital pressure sensor has been designed and installed under the prism, to ensure that no deformation occurs under the action of the pressure. The sensor array displays the value of pressure in real time, without changing the position of the prism. Figure 2a shows the schematic diagram of the digital pressure sensor array device, whereas Figure 2b shows the corresponding setup. The sensor is composed of five parts: connection block, pressure sensor, protective film, transmission device, and digital display.

### 2.2. Experimental Processing

In order to improve precision of in vivo THz measurements, the humidity in the sealing cover was less than 5% through nitrogen in a chamber. Based on this, the reference signal was obtained with air. The volunteers pushed his/her thumb on the prism surface under different conditions (different thumb/prism relative positions, different contact pressure, different contact durations, and different materials of prism) to obtain reflection spectra. We controlled variables to obtain time domain spectra during the processing. It is crucial to get favourable optical contact between the thumb and the prism. Glycerol solution (0.2 ml 84% glycerol) was used to improve this optical contact for 10 min, thus increasing the penetration depth into the thumb. After each experiment, the prism needs to be cleaned by alcohol. The measurements were carried out on three volunteers. They were in good health and their thumbs were not injured. The volunteers signed an informed consent agreement. They could leave the study at any time and did not have to give a reason.

### 2.3. Data Processing

In this section, we describe our method in some detail to determine the THz absorption and the refractive index of skin on the surface of the thumb. In the THz-TDS framework, time-domain measurements are performed, providing both amplitude and phase information from the sample and a reference. The frequency spectrum may be obtained from the time-domain signal via Fourier Transform, and then used to recover the absorption rate and the complex refractive index. This is done by calculating the reflection coefficients of the sample and the reference using the Fresnel Equations [30,44,45]. We thus obtained a reflection coefficient ratio between the sample and a reference (air). Upon denoting the detected signal with the thumb on the prism by $E_{\mathrm{sample}}$ and the signal without thumb on the prism by $E_{\mathrm{reference}}$, the THz reflectivity can be defined as [39,46]:(1)$$r=\frac{FFT(E_{\mathrm{sample}}\left(t\right))}{FFT(E_{\mathrm{reference}}\left(t\right))}=\frac{r_{\mathrm{sample}}}{r_{\mathrm{reference}}}$$
where $r_{\mathrm{reference}}$ is the Fresnel reflection coefficient with thumb on the prism and $r_{\mathrm{sample}}$ is the Fresnel reflection coefficient without thumb on the prism, which may be expressed as:(2)$$r_{\mathrm{sample}}=\frac{n_{\mathrm{sample}}\cos\theta -n_{\mathrm{prism}}\cos\gamma_{\mathrm{sample}}}{n_{\mathrm{sample}}\cos\theta +n_{\mathrm{prism}}\cos\gamma_{\mathrm{sample}}}$$
(3)$$r_{\mathrm{reference}}=\frac{n_{\mathrm{air}}\cos\theta -n_{\mathrm{prism}}\cos\gamma_{\mathrm{air}}}{n_{\mathrm{air}}\cos\theta +n_{\mathrm{prism}}\cos\gamma_{\mathrm{air}}}$$

In Equations (Equation 1) and (Equation 2)
(4)$$\cos\gamma_{\mathrm{sample}}=\sqrt{1-{\left(\frac{n_{\mathrm{prism}}}{n_{\mathrm{sample}}}\sin\theta \right)}^{2}}$$
(5)$$\cos\gamma_{\mathrm{air}}=\sqrt{1-{\left(\frac{n_{\mathrm{prism}}}{n_{\mathrm{air}}}\sin\theta \right)}^{2}}$$
where θ is angle of incidence, γ is angle of reflection, $n_{\mathrm{sample}}$, $n_{\mathrm{prism}}$, $n_{\mathrm{air}}$ represents the refractive indices of sample, prism, and air, respectively. By substituting Equations (Equation 2)–(Equation 5) into Equation (Equation 1), the complex dielectric function $\epsilon_{\mathrm{sample}}$ may be written as: $$\epsilon_{\mathrm{sample}}={n_{\mathrm{sample}}}^{2}$$
$$={n_{\mathrm{prism}}}^{2}\frac{{\left(1+r\cdot r_{\mathrm{reference}}\right)}^{2}\pm \sqrt{{\left(1+r\cdot r_{\mathrm{reference}}\right)}^{4}-4{\left(1-r\cdot r_{\mathrm{reference}}\right)}^{2}{\left(1+r\cdot r_{\mathrm{reference}}\right)}^{2}{\left(\sin\theta \right)}^{2}{\left(\cos\theta \right)}^{2}}}{2{\left(1-r\cdot r_{\mathrm{reference}}\right)}^{2}{\left(\cos\theta \right)}^{2}}$$
(6)$$=\epsilon^{\prime }-i\epsilon^{\prime \prime }$$
where $\epsilon^{\prime }$ and $\epsilon^{\prime \prime }$ represent the real and the imaginary part of the complex permittivity of the thumb. The refractive index $n_{\mathrm{sample}}$ is given by:(7)$$n_{\mathrm{sample}}=\sqrt{\epsilon_{\mathrm{sample}}}=n-ik$$
where *n* and *k* represent the real and the imaginary part of the refractive index of the thumb. By using Equations (Equation 6) and (Equation 7), the optical constants of the skin surface may be extracted and analysed.

### 2.4. THz Response with Different Thumb/Prism Relative Positions

In our experiment, a volunteer is asked to place his/her thumb on the top of the prism (silicon) surface. However, during the measurement, the exact position of the thumb affects the actual measurement. Therefore, in order to verify the position of the contacting point on the prism and to find the optimal one, we have performed measurements using different regions of the prism. Figure 3 shows three regions of the prism.

Taking the centre of the prism as the origin, parallel to the length as X-axis and parallel to the width as Y-axis, we evenly divide the prism into three regions: region 1, region 2 and region 3. Free area for sample placement is 2.5 cm (width) × 6.2 cm (length). The central coordinates of regions 1, 2, 3 is (–2.1,0), (0,0), (2.1,0), respectively. The volunteer’s thumb is placed on the region 1, region 2, and region 3, respectively, and then measured by the THz-ATR spectroscopy system. In Figure 4a, the reference signal is denoted by a solid black curve. This corresponds to the signal obtained with air when the humidity in the sealing cover is lower than 5%. The pressure applied to the prism by the thumb is 1 N. The time domain signals measured at different regions are shown in Figure 4a. Data are obtained by performing repeated measurements for three volunteers in succession for each region.

The waveforms and amplitudes of the time domain signals measured with the thumb placed on region 2 of the prism are significantly lower than those obtained with the thumb on region 1 and 3. The time domain signals with the thumb on region 1 and region 3 are slightly different compared with the reference signal (air), whereas waveforms are basically unchanged and the amplitudes are slightly reduced. In Figure 4b, we report the reflectivity in the frequency domain, which is obtained by Fourier transform of the time domain signals. Figure 4c shows the error tree diagram of the reflectance at 0.3 THz obtained by repeated measurements of the thumb placed in three regions of the prism surface.

Since experiments are conducted in unsealed conditions, water vapour may be presented in the measurement area, and the absorption peaks of water appears at the individual frequency points. Our results show that at fixed thumb pressure, the absorption with the thumb placed on region 2 of the prism is stronger than that measured with the thumb on region 1 and 3. This suggests that the THz wave is better reflected through the prism surface when the thumb is placed on region 2, which represents the optimal choice for this kind of in vivo measurement.

### 2.5. THz Response with Different Contact Pressure

The contact between the thumb and the prism surface leads to mechanical deformation of the surface of the skin. This, in turn, implies that pressure affects the THz response of skin due to skin hydration. In order to improve precision of the measurements, we study how contact pressure affects the THz response during in vivo THz measurements on skin. To this aim, and in order to control the pressure between the thumb and the prism, a digital pressure sensor has been designed and installed under the prism as in Figure 2.

As mentioned above, the reference signal (air) is taken when the humidity in the sealing cover is less than 5 %. Then the thumb is placed on region 2 of the prism and time domain measurements are taken by increasing pressure by 0.1 N at each step. Results are shown in Figure 5a. Compared to the reference signal (air), the waveform changes significantly after the thumb is put in contact with the prism. By taking multiple measurements at different pressures, we have built the error bar graph of Figure 5b between the peak and the valley (ΔA). As the contact pressure increases, the amplitude of the time domain signal decreases.

The Fourier transforms of the time domain signals, i.e., the reflectivities in the frequency domain, are shown in Figure 5c, whereas Figure 5d reports an error tree diagram of the reflectance at 0.3 THz obtained by multiple measurements of the thumb placed on region 2 of the prism with different pressures. Several absorption peaks may be observed in the reflection spectra, due to ambient water vapour and the high sensitivity of THz radiation to water concentration. It is apparent from Figure 5d that the reflectivity decreases with the increase of pressure between the thumb and the prism. This indicates that after the thumb is placed in contact with the prism, water starts to accumulate on the surface of the skin, thus changing the reflectivity of the tissue and increasing the absorption of the THz wave. In Figure 5e, we show the refractive index spectra of the thumb skin for different contact pressure, which is calculated from the reflection spectra using Fresnel equations, whereas Figure 5f contains the corresponding error tree diagram of the reflectance at 0.3 THz. The complex permittivity of skin can be described by the permittivity of effective dry tissue and water, depending on the volume concentration of water solution [25,31,39]. Thus, the imaginary part of the permittivity of skin represents the water content of the skin surface. It is noted that for increasing pressure, the skin surface is compressed and therefore water concentration and biological density increase. In summary, it is demonstrated that the THz response of skin is very sensitive to contact pressure, which further increases the refractive index of the thumb skin and the water content of the skin surface.

In summary, our results show that contact pressure is a relevant factor in THz ATR experiments. Upon controlling the contact pressure, one would eliminate the disturbance caused by the change of the water concentration of the skin, thus improving the overall stability and accuracy of the technique.

### 2.6. THz Response with Different Contact Durations

If the thumb is in contact with the prism, the pressure causes water to accumulate on the surface of the skin, thus changing the water concentration of the skin itself. Therefore, the duration of the contact between the thumb and the prism should be taken into account as a relevant factor affecting the accuracy of the ATR-THz technique.

In analogy with the procedure adopted in previous Sections, we start by reducing the humidity in the sealing cover below 5%, then place the thumb on region 2 of the prism and carefully control the pressure, which is maintained at 0.5 N during data acquisition. We then take measurements at one minute intervals without changing the position of the thumb and the pressure. The THz-ATR time domain spectra of after 5, 10, 15, and 20 min of continuous contact of the thumb are shown in Figure 6a. The waveform is almost unchanged, whereas the amplitude slightly changes. The peak-to-peak amplitude between the peak and the valley (ΔA) is shown in Figure 6b. It is clear that the greatest change in the THz signal occurs in the first two min and that after three min, the reduction rate of the THz signal is smaller.

The Fourier transform of the time domain signal is reported in Figure 6c, whereas the error tree diagram of the reflectance at 0.6 THz obtained by multiple measurements is shown in Figure 6d. As the contact duration increases, the reflectivity decreases continuously, that is, the water content of the skin surface increases with time. The refractive index of the thumb skin is reported in Figure 6e, whereas Figure 6f shows the error tree diagram of the refractive index at 0.6 THz, which is obtained by multiple measurements. The refractive index of the thumb increases with time, and this means that measurements should be performed when the accumulation of water is stabilised.

### 2.7. THz Response with Prism Made of Different Materials

Attenuated total reflection of THz waves occurs at the interface. The penetration depth of the evanescent wave is limited and exponentially decays, we thus employ prisms of different material (Teflon and silicon) and very different refraction indices to assess the corresponding effect on the ATR output signal. The gray black prism in Figure 7a is a silicon prism with a refractive index of 3.42, whereas the white one in Figure 7b is a Teflon prism with a refractive index of 1.446. In order to properly compare experimental results, we have performed measurements with the two prisms using the same person’s thumb placed in the same position and with the same contact duration. The measured time domain spectra with pressures of 0.2, 0.3, and 0.4 N are shown in Figure 7a,b. The signal detected with the Teflon prism is less pronounced than the silicon one. This may be explained by taking into account that, in this case, non-total reflection occurs and most of the energy is transmitted to the human skin at the interface between the Teflon prism and the thumb, such that the amplitude of the time domain signal significantly decreases. In addition, for increasing pressure, the amplitude of the time domain signal decreases, due to the increase of water concentration on the surface of the skin.

The Fourier transforms of the time domain signals are shown in Figure 7c,e, whereas the error tree diagrams of the reflectance at 0.3 THz are shown in Figure 7d,f. Upon comparing Figure 7d,f, one sees that the reflectivity at 0.3THz decreases with pressure: the reflectivity measured with the Teflon prism is much smaller than that obtained with the silicon prism, in agreement with the different detection results of non-total reflection and total reflection. It is noted that there are three peaks in Figure 7c because the experimental environment is open and the humility will increase as time goes by. Even so, these three sharp resonances have little influence on the results because the skin shows no specific fingerprint resonance.

The refractive index of the thumb skin is calculated by the reflection spectrum and shown in Figure 8a,c. By comparing Figure 8b,d, one sees that two refractive indices are different in value. This may be explained that the depth of the THz wave permeating the skin of the thumb is different, and the refractive index of the actual human skin is not uniform with the increasing of the skin depth. In other words, although the THz wave may penetrate deeper into the skin when the Teflon prism is present, the uneven refractive index of the skin causes interference and the overall effect is detrimental.

### 2.8. Systematic and Random Errors

For the commonly used ATR transmission geometry, the fact that the prism is not firmly fixed on to the platform, which causes small yet non-negligible variation in the position or angle of the prism, gives rise to the systematic error. To avoid systematic error, we use a sample holder for ATR measurements, which is designed and produced by BATOP GmbH, Stockholmer Str. 14, D-07747 Jena, Germany. The prism is stably fixed on the optical platform. In addition to systematic errors, the random errors also affect the experimental results, which may arise from laser power fluctuations, electrical noise in the electronic circuitry, and delay jitter. Hence, we use ultrafast pump laser femto fibre pro NIR (Toptica Inc., Munich, Germany) in a temperature (within 22–23 °C) and humidity (below 5%) environment (void of dust and aerosols), digital signal processing (DSP) card integrated into a TMS-200A electronic module, and scanning fast optical delay line in 10 Hz frequency. To further avoid the random error and systematic error, we use multiple measurements to reduce possible errors. Other approaches can reduce or eliminate systematic error, for example, using a long prism fixed to an optics translation stage [19]. In our future work, we may use a similar long prism for further systematic error improvement.

## 3. Potential Applications

### 3.1. THz Response to Different Handcream

The handcream is generally applied to prevent hand skin from cracking. THz ATR configuration can be utilised to estimate water motion in human skin, thus assessing the properties of handcream.

In order to properly compare skin properties, two different types of handcream are selected. Their ingredients and functions are listed in Table 1. In the experiment, handcream with 2 mL volume was extracted into the needle tube, as shown in Figure 9a. In combination with the lists of ingredients, handcream 1 is transparent and moisturising, in turn, handcream 2 has thick texture and locks a strong moisture, as shown in Figure 9b. When handcream works with the human skin via silicon prism, it can be approximated as the stratified media model including three main parts: stratum corneum (SC), epidermis, dermis [25,47,48], as shown in Figure 9c.

When the humidity in the sealing cover is lower than 5%, the same person’s thumb is placed on region 2 of the silicon prism by 1 N for 3 min. The measured time domain signal without handcream is taken as reference in Figure 10a. We then apply two handcreams on the thumb uniformly and use a slide on the contact area for a while to fully absorb. The corresponding time domain spectra are shown in Figure 10a, whereas waveforms are basically unchanged and the amplitudes are slightly reduced. The Fourier transforms of the time domain signals are reported in Figure 10b. We see that the reflectivity of handcream 1 is lower than handcream 2, in turn, the refractive index of the thumb skin increases. This may be explained by the fact that handcream 1 is watery, causing the water content of the skin surface to increase, thus increasing the absorption of the THz wave.

### 3.2. Theory and Simulation

To substantiate properties of different handcream, we performed 2D numerical simulations using commercial software, COMSOL Multiphysics. THz wave with TE polarisation ($\hat{E}$ in the $\hat{z}$ direction) is incident on the prism/air interface at an angle θ of 52. The y-plane is set as the Floquet boundary. A waveguide port is set as the periodic port, simulating a transverse magnetic plane wave incident. The dielectric permittivity of each layer is from reference [48]: $\epsilon_{SC}$ = 2.7 (100 μm), $\epsilon_{epidermis}$ = 3.25 (600 μm), $\epsilon_{dermis}$ = 3.9. It is generally recognised that the most component of handcream is water, thus the dielectric permittivity of handcream is approximate to water. In the THz range, the double Debye mode is used to represent the permittivity of water [49,50].

For the TE-polarised terahertz wave beam, the reflection coefficients (*r*) of the proposed structure can be expressed as follows [51]
(8)$$r=\frac{r_{01}+r_{1234}e^{i2k_{z1}d_{1}}}{1+r_{01}r_{1234}e^{i2k_{z1}d_{1}}}$$
where $r_{01}=\left(k_{0z}-k_{1z}\right)/\left(k_{0z}+k_{1z}\right)$, $r_{12}=\left(k_{1z}-k_{2z}\right)/\left(k_{1z}+k_{2z}\right)$, $r_{23}=\left(k_{2z}-k_{3z}\right)/\left(k_{2z}+k_{3z}\right)$, $r_{34}=\left(k_{3z}-k_{4z}\right)/\left(k_{3z}+k_{4z}\right)$, $r_{234}=\left(r_{23}+r_{34}e^{i2k_{3z}d_{3}}\right)/\left(1+r_{23}r_{34}e^{i2k_{3z}d_{3}}\right)$, $r_{1234}=\left(r_{12}+r_{234}e^{i2k_{2z}d_{2}}\right)/\left(1+r_{12}r_{234}e^{i2k_{2z}d_{2}}\right)$, $k_{zi}=\sqrt{{k_{i}}^{2}-{\left(k_{1}\sin\theta \right)}^{2}}$ is the *z*-component of wave vector in medium *i* (θ: the incident angle of the terahertz wave), $k_{i}$ is the wave number of light in medium *i*, $k_{i}=n_{i}k_{0}$, $n_{i}$ is the refractive index of medium *i* (*i* = 0, 1, 2, 3, 4), $d_{1}$, $d_{2}$, $d_{3}$ is the thickness of the handcream, SC layer, epidermis layer, respectively.

By numerical calculation, we show in Figure 10b how the reflectivity of handcream changes with the frequency. The reflectivity of handcream 1 is smaller than handcream 2, which is in good agreement with the experimental results. The two curves experience ups and downs and tend to be saturated around 0.55 THz. In this way, it is possible to quantitatively evaluate the properties of handcream.

## 4. Conclusions

In this paper, we have systematically investigated the effects of several experimental factors on the THz response of in vivo THz-TDS spectroscopy on human skin by an ATR optical scheme. In particular, we have discussed in detail the influence of the contact positions between the thumb and the prism, the contact pressure, the contact duration, and the materials of the prism. Our results prove that an optimal configuration is achieved by placing the thumb on region 2 of a silicon prism. In addition, we have seen that by increasing the contact pressure and the contact duration, the amplitude of the ATR signal decreases and the refractive index of the thumb skin increases. In practice, we apply the stratified media model to quantitatively evaluate and develop different types of handcream. Our results make it clear that all the variables should be carefully controlled, which pave the way for applications of ATR based THz-TDS spectroscopy in realistic medical treatments.

## Figures and Tables

**Figure 1 sensors-20-04256-f001:**
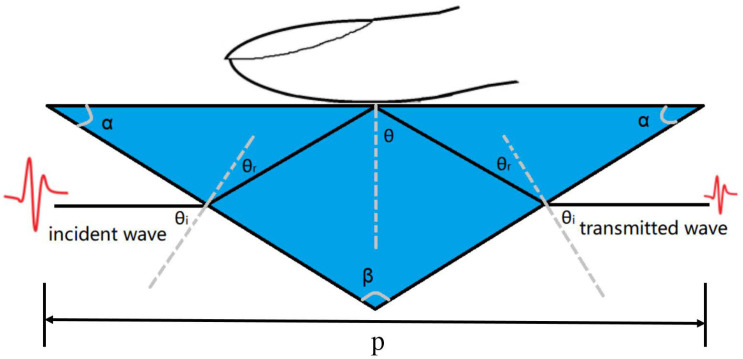
Schematic diagram of the prism ($n_{\mathrm{prism}}$ = 3.42): the bottom length *P* = 6.2 cm, the top angle β = 104, the bottom angle α = 38, angle from the perpendicular line θ = 51.6, contacting angle between prism and sample $\theta_{i}$ = 52, entering the prism refraction angle $\theta_{r}$ = 13.3.

**Figure 2 sensors-20-04256-f002:**
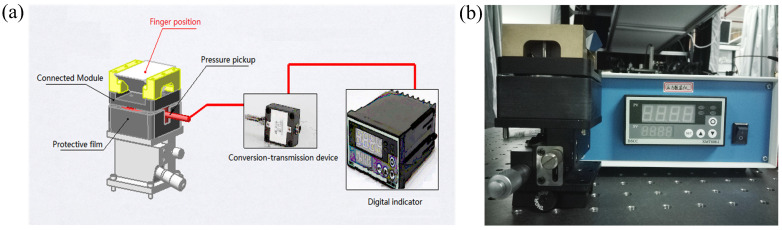
(**a**) The schematic diagram of the digital pressure sensor array device. (**b**) The setup of the digital pressure sensor array device.

**Figure 3 sensors-20-04256-f003:**
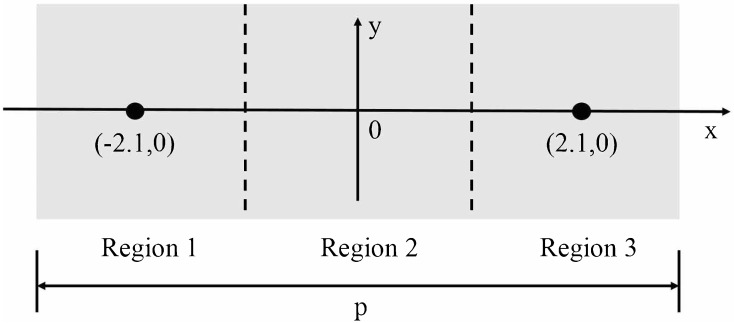
The volunteer’s thumb is placed at different regions on the surface of the prism: region 1–3. Free area for sample placement is 2.5 cm (width) × 6.2 cm (length).

**Figure 4 sensors-20-04256-f004:**
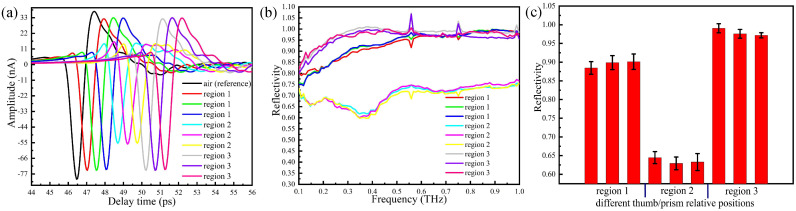
Terahertz (THz)-attenuated total reflection (ATR) spectra with the thumb placed at different regions on the prism: (**a**) the time domain spectra, (**b**) the reflection spectra, (**c**) error tree diagram of the reflectance at 0.3 THz.

**Figure 5 sensors-20-04256-f005:**
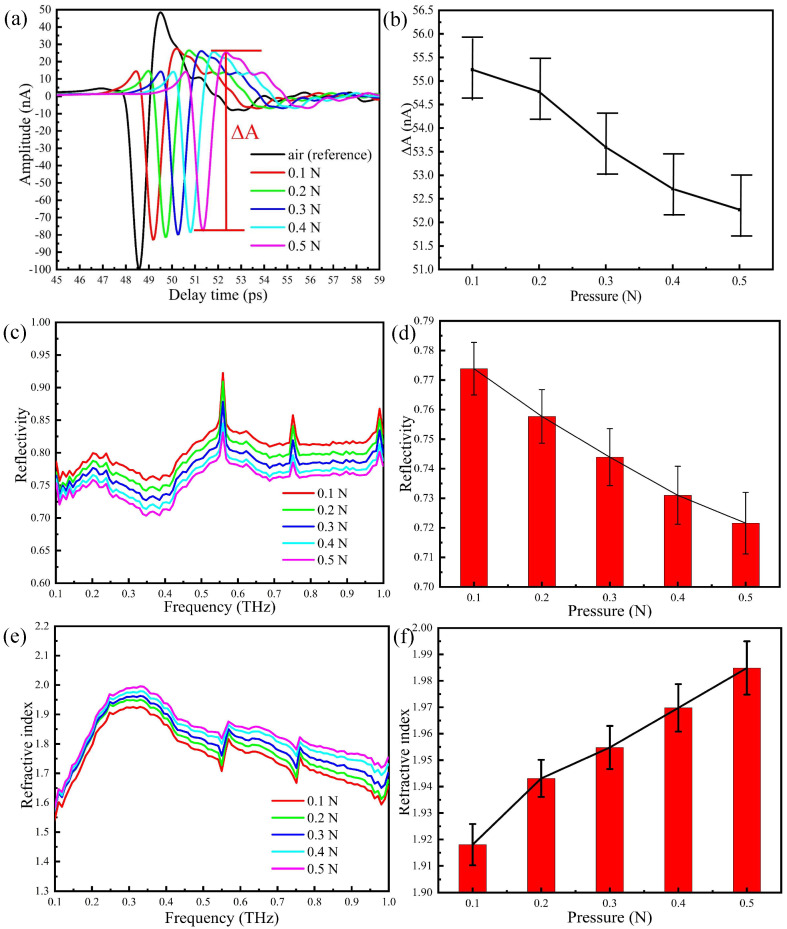
THz-ATR spectra obtained by placing the thumb on region 2 of the prism and with different pressures: (**a**) the time domain spectra, (**b**) the peak-to-peak amplitude, (**c**) the reflection spectra, (**d**) the error tree diagram of the reflectance at 0.3 THz, (**e**) the refractive index spectra, (**f**) the error tree diagram of the reflectance at 0.3 THz.

**Figure 6 sensors-20-04256-f006:**
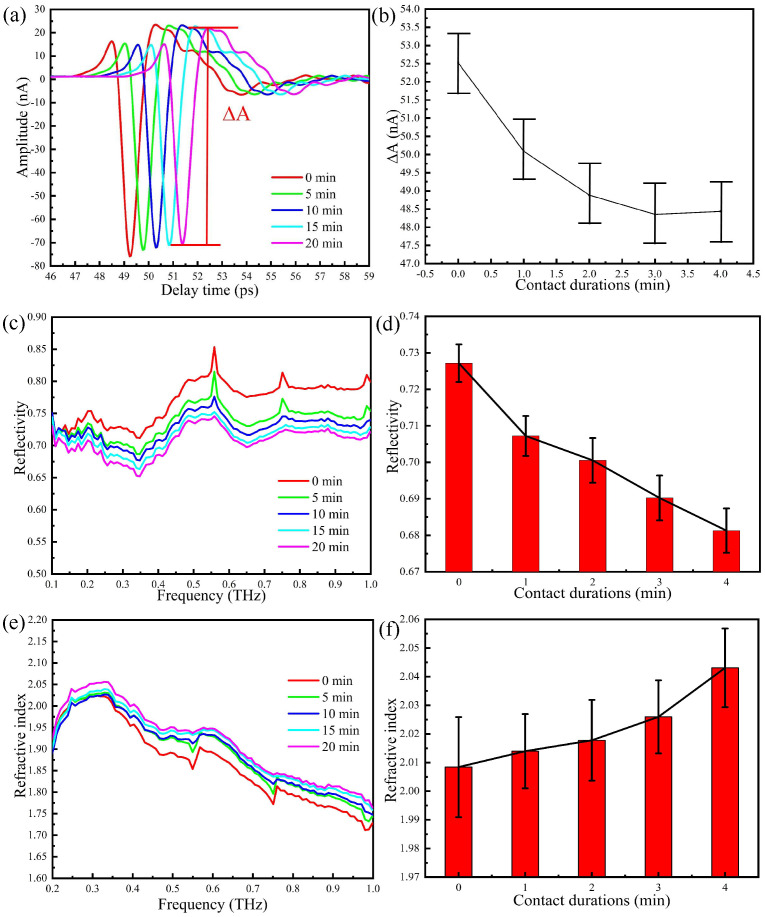
THz-ATR spectra with the thumb on region 2 of the prism and different durations of the contact between the thumb and the prism: (**a**) the time domain spectra, (**b**) peak-to-peak amplitude, (**c**) the reflection spectra, (**d**) error tree diagram of the reflectance at 0.6 THz, (**e**) the refractive index spectra, (**f**) error tree diagram of the reflectance at 0.6 THz.

**Figure 7 sensors-20-04256-f007:**
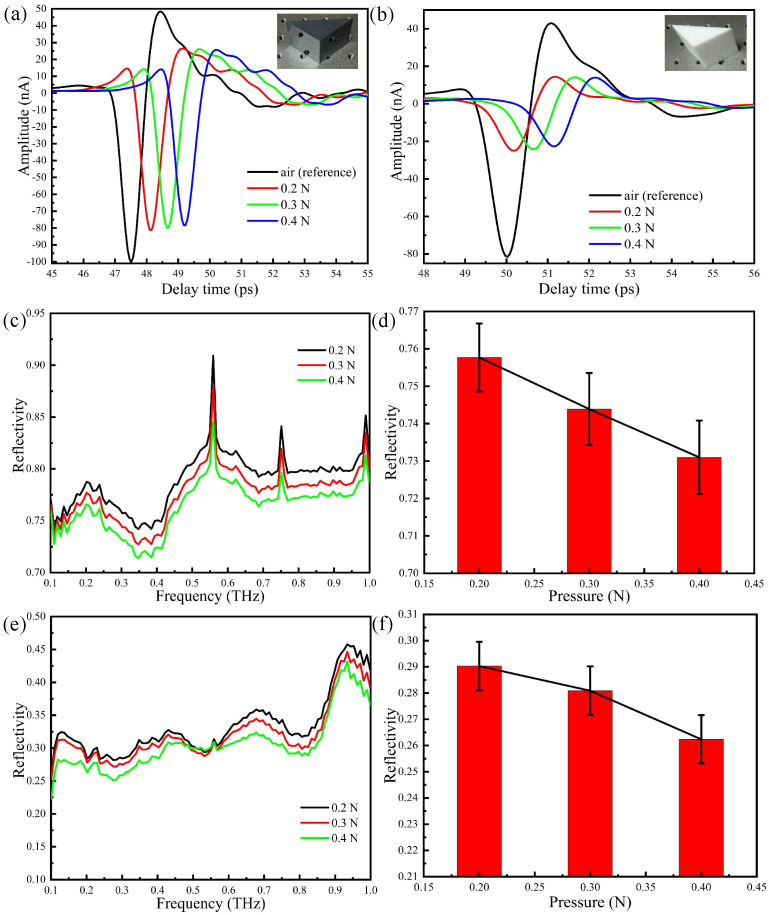
THz-ATR spectra with different different prisms: (**a**) the time domain spectra: silicon; (**b**) the time domain spectra: Teflon; (**c**) the reflection spectra: silicon; (**d**) error tree diagram of the reflectance at 0.3 THz: silicon; (**e**) the reflection spectra: Teflon; (**f**) error tree diagram of the reflectance at 0.3 THz: Teflon.

**Figure 8 sensors-20-04256-f008:**
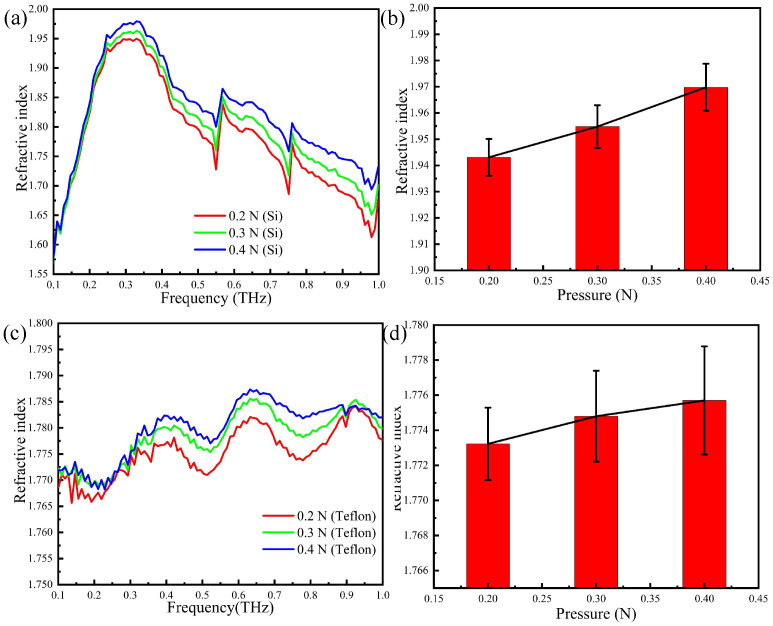
The refractive index of the thumb skin with different prisms: (**a**) the refractive index spectra: silicon; (**b**) error tree diagram of the reflectance at 0.3 THz: silicon; (**c**) the refractive index spectra: Teflon; (**d**) error tree diagram of the reflectance at 0.3 THz: Teflon.

**Figure 9 sensors-20-04256-f009:**
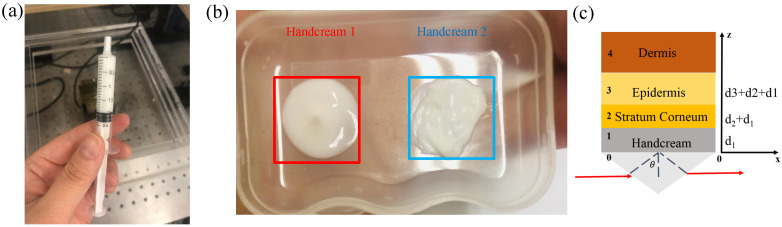
(**a**) 2 mL handcream; (**b**) textures of two types of handcream; (**c**) schematic illustration of the stratified media model.

**Figure 10 sensors-20-04256-f010:**
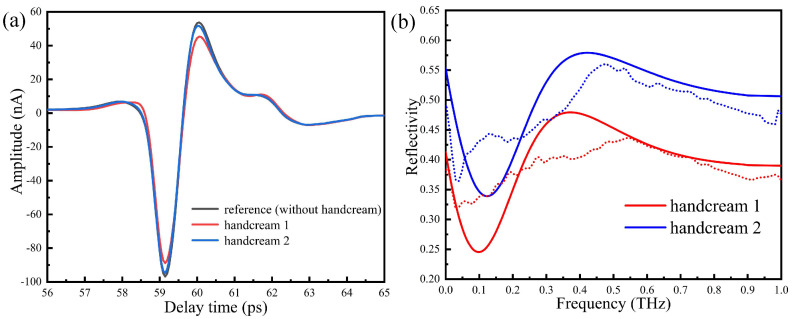
(**a**) The time domain spectra, (**b**) the reflection spectra. Dotted lines are experiment results and solid lines are calculated results.

**Table 1 sensors-20-04256-t001:** The major ingredients of handcream 1, 2.

Handcream	Handcream 1				Handcream 2			
Ingredients	Aqua	Glycerol	Caprylic/ Capric Triglyceride	Sorbitan Oleate	Aqua	Isopropyl Myristate	Huile minerale	Glyceryl Stearate
Function	Solvent	Moisturize	Parfum	Emulgator	Solvent	Ester	Solvent	Emulgator

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
