# Peer review of "Critical Factors for In Vivo Measurements of Human Skin by Terahertz Attenuated Total Reflection Spectroscopy"

_sensors, 2020, doi:10.3390/s20154256_

Round 1

Reviewer 1 Report

Terahertz (THz) spectroscopy and imaging have potential for human diagnosis in the future. THz-TDS combined with ATR is a good choice for in vivo skin analysis. This paper discussed the influence of several factors and optimization of some parameters. The results given are helpful for readers who apply THz-TDS for skin and cancer diagnosis.

Some comments and revision suggestions are given below:

  1. In section 3.1, the positions of thumb on the prism were shown in the Figure 4. However, the words left, middle, and rightare qualitative description,. Please give a quantitative measure, such as a distance in centimeter.  
  2. In section 3.3. THz response with different contact durations, as shown in the Figure 6(b),showing that the amplitude of the instantaneous time domain signal decreases continuously with time, nearly achieving a stationary value after 3 minutes. Therefore, it is concluded that the amplitudereaches its saturation value after 3 minutes. However, from Figure 6(d), we it can be seen that the reflectivity decreases until the test range end (4.0 min), and from the Figure 6(f), the refractive index keeps increasing to the test end (4.0 min). The results of Figure 6(d) and 6(f) seem conflicting with that shown in the Figure 6(b). Please explain.  
  3. The comparison of prism materials is interesting. Although the signal with silicon prism is higher than that of using Teflon prism, three sharp peaks were measured in the range of 0.1-1.0 THz. This means that these peaks may cause serious interference in practical applications. However, the no sharp peak was observed when using Teflon prism, so it may have less spectral interference in practical measurement. Please give an explanation.
  4. On page 7, line 158, ⋅⋅⋅and functions are listed in Table 1 and 2.”. However, only Table 1 was found in the manuscript. Please give the Table 2.

Reviewer 2 Report

Dear Authors, 

Please find the attached comments and suggestions. 

Round 2

Reviewer 2 Report

Dear Authors

Many thanks for updating the manuscript. 

Best Wishes